# Novel Biotransformation of Maslinic Acid to MA-2-O-β-D-Glucoside by UDP-Glycosyltransferases from *Bacillus subtilis*

Fen Hu [1], Jiaxin Chen [1], Yunfeng Zhang [2,*], Yuxi Sun [1], Yan Liu [1], Yuan Yu [1], Ke Xu [2,3,*] and Haifeng Cai [4,*]

[1] College of Life Sciences, North China University of Science and Technology, Tangshan 063210, China
[2] Tangshan Key Laboratory of Agricultural Pathogenic Fungi and Toxins, Department of Life Sciences, Tangshan Normal University, Tangshan 063000, China
[3] Key Lab for Industrial Biocatalysis, Ministry of Education, Department of Chemical Engineering, Tsinghua University, Beijing 100084, China
[4] The Second Department of Breast Surgery, Tangshan People's Hospital, Tangshan 063000, China
* Correspondence: yunfengzhang1982@126.com (Y.Z.); xuke528@tsinghua.edu.cn (K.X.); 13303050005@163.com (H.C.)

**Abstract:** Maslinic acid (MA) is a pentacyclic triterpenoid which originates from olive and other plants. Though MA possesses multiple biological activities, it has limitations due to its poor water solubility. YojK, YjiC, and UGT109A3 UDP-glycosyltransferases (UGTs) from *Bacillus subtilis* (*B. subtilis*) were utilized to catalyze the conjugation of MA with UDP-Glucose to generate a new MA glycosylation product, MA-2-O-β-D-glucoside (MA-2-O-β-D-Glu). The experimental results indicated that the resultant water solubility of MA-2-O-β-D-Glu is 1.69 times higher than that of MA. In addition, the recombinant YojK showed maximum activity at 40 °C with a pH range of 8.0−10.0, while the recombinant YjiC showed maximum activity at 45 °C with a pH of 8.0, and the recombinant UGT109A3 showed maximum activity at 40 °C with a pH of 8.0. $Mg^{2+}$ is an important factor for efficient catalysis by three recombinant glycosyltransferases. The chemical conversion rate of the recombinant YojK, YjiC, and UGT109A3 is nearly 100% at their optimum pH, temperature, and metal ions. Furthermore, eight essential residues of three UGTs for MA glycosylation modification were further determined by molecular docking and site-directed mutagenesis. Thus, efficient glycosylation modification improves the water solubility of MA and provides a new potential method for the glycosylation modification of other pentacyclic triterpenoids.

**Keywords:** maslinic acid; glycosylation; biocatalysis; UDP-glycosyltransferases

## 1. Introduction

Pentacyclic triterpenoids (PTs) are a class of compounds that widely exist in nature and have a wide range of medical applications [1]. Moreover, the hydroxyl, carboxyl, and glycosyl functional groups on the skeletons of PTs are essential for their biological activity [2]. Glycosylation modifications are known to contribute to the structural diversity of various PTs. Maslinic acid (MA) belongs to oleanane PTs and has comprehensive medicinal value, including anti-cancer [3], anti-inflammatory [4], anti-bacterial [5], and anti-HIV [6].

Glycosylation modification is an essential biological mechanism for increasing the biological activity of natural products. Glycosylation modification can be accomplished by UDP-glycosyltransferases (UGTs), whether in natural biological systems or in vitro synthetic systems. UGTs catalyze the glycosylation modification of various natural products, such as crocin [7], pterostilbene [8], flavonoids [9], ginsenosides [10], and glycyrrhetinic acid (GA) [11–13]. Recently, YjiC has been reported to aid the transformation of Ginsenoside Rg3 (a PPD-type ginsenoside) into Ginsenoside Rd12 [14]. GA glycol-diversification by YojK [12] and UGT109A3 [13] UDP-glycosyltransferase have also been reported previously.





MA is a good candidate for glycosylation given the presence of the activated C2-hydroxyl, C3-hydroxyl, and C28-carbonyl groups. To enhance the bioactivities and medicinal value of MA, different MA-glycosides can be synthesized by enzymatic glycosylation. In this study, we investigate UGTs, which have catalytic activity for the biotransformation of MA to MA-glycosides. MA-glycosides may exhibit improved water solubility and potential activity.

In this study, three UGTs (YojK, YjiC, and UGT109A3) from *B. subtilis* were recombinantly expressed and investigated for their ability to glycosylate MA in vitro (Figure 1A). In addition, optimum reaction conditions of these UGTs were analyzed. Furthermore, the critical amino acid sites of three glycosyltransferases were identified. Moreover, the water solubility of MA and MA-2-O-β-D-Glu was determined.

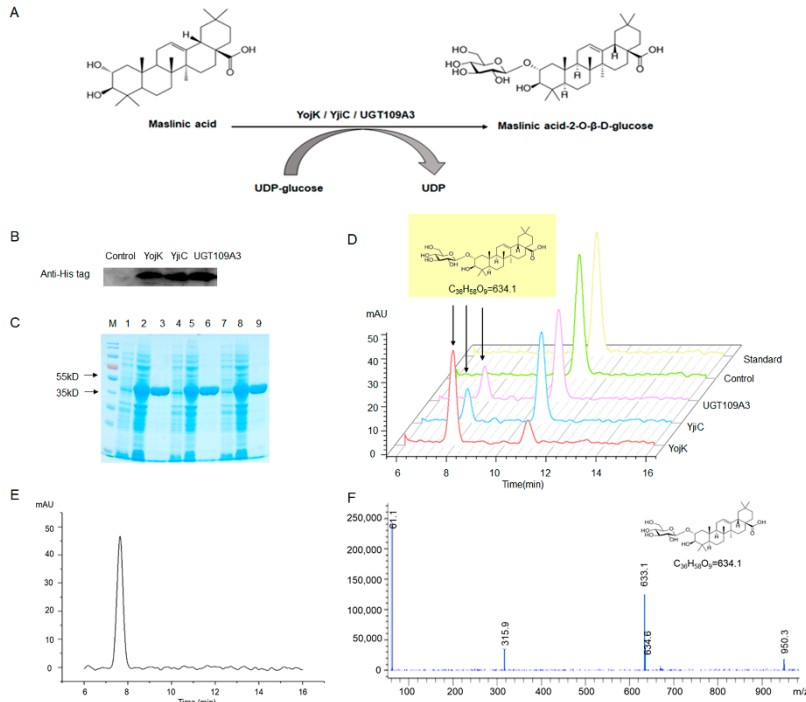

**Figure 1.** Enzymatic glycosylation of MA with YojK, YjiC, and UGT109A3. (**A**) The glycosylation schematic of MA to MA-2-O-β-D-Glu by YojK, YjiC, and UGT109A3. Western blot (**B**) and SDS-PAGE (**C**) analysis of YojK, YjiC, and UGT109A3 expression. (**D**) HPLC analysis of MA glycosylation by YojK, YjiC, and UGT109A3. M: protein marker; lane 1, 4, and 7: control; lane 2, 5, and 8: YojK, YjiC, and UGT109A3 cell lysate respectively; lane 3, 6, and 9: Purified YojK, YjiC, and UGT109A3, respectively. HPLC (**E**) and LC-MS (**F**) analysis of purified MA-glycoside product.

## 2. Results and Discussion

### 2.1. Expression and Purification of YojK, YjiC, and UGT109A3

In order to increase the water solubility and bioactivities of MA, glycosylation of MA was catalyzed by UGTs. It has been reported that YojK and UGT109A3 can catalyze Glycyrrhizic Acid (GA) glycosylation [11–13]. Therefore, based on literature precedent, we made a similar assumption for MA, given its structural similarities to GA. First, three UGTs (YojK, YjiC, and UGT109A3) expression plasmids were constructed, and their expression was confirmed by Western blot (Figure 1B) and SDS-PAGE (Figure 1C). Then, recombinant YojK, YjiC, and UGT109A3 were purified and then analyzed by SDS-PAGE (Figure 1C). Clear bands at 35–55 kDa (over 90% purity) were shown by SDS-PAGE analysis, which was consistent with the calculated molecular weight of the recombinant YojK (45.6 kDa), YjiC (46 kDa), and UGT109A3 (43.7 kDa), respectively.

### 2.2. Enzymatic Activities of UTGs for MA Glycosylation

HPLC was used to analyze the MA glycosylated derivatives, which were catalytically produced by YojK, YjiC, and UGT109A3 enzymes. Retention peak for MA appeared at retention time (RT) of 10.7 min for control and MA standard, and one new retention peak appeared at RT of 7.8 min for the recombinant YojK, YjiC, and UGT109A3, compared with control reactant (Figure 1D). This indicates that one new product was produced by recombinant YojK, YjiC, and UGT109A3.

### 2.3. Identification of MA Glycosylation Products

To analyze the chemical structure of MA glycosylation derivatives, MA glycosylation derivatives were purified with preparative HPLC; the isolated yields of MA catalyzed by YojK, YjiC, and UGT109A3 were 52%, 40%, and 39%, respectively. Purified MA glycosylation derivatives appeared at 7.8 min in HPLC analysis (Figure 1E). Then, the LC-MS analysis result of the purified compound indicated that YojK, YjiC, and UGT109A3 transferred one glucose moiety ($C_6H_{12}O_6$) to MA ($C_{30}H_{48}O_4$) to yield a $C_{36}H_{58}O_9$ compound (Figure 1F). Furthermore, its structure was elucidated by the $^1H$ and $^{13}C$ in the NMR spectrum (Table 1). The $^{13}C$ NMR spectrum of the new product exhibited six new signals at $\delta$ 102.82, 73.80, 77.00, 70.41, 77.18, and 61.42 ppm, which were characteristic signals of D-glucose. Moreover, the anomeric proton signal H1' ($J$ = 7.8 Hz) indicated that the C2 glucosyl moiety of MA adopted the β-configuration. The substitution of a glucose at position C-2 was further confirmed by $\delta$H 3.55 and $\delta$C 79.44. Thus, the structure of this new compound, an MA-derived unnatural product, was MA-2-O-β-D-Glu (Table 1). The C3 hydroxyl and C30 carboxyl groups of GA are the active sites, which can be glycosylated by UGT109A3, UGT109A3, and YojK [12,13]. In addition, the active site C3 hydroxyl group of oleanolic acid can be glycosylated by *B. subtilis* ATCC 6633 [15]. MA has free C2 hydroxyl, C3 hydroxyl, and C28 carboxyl groups, but YojK, YjiC, and UGT109A3 only glycosylate the free C2 hydroxyl group of MA to yield a unique MA-2-O-β-D-Glu. Furthermore, the experimental results showed that the water solubility of MA-2-O-β-D-Glu was 1.69 times that of MA. MA-2-O-β-D-Glu may be a potential leading compound due to its higher water solubility.

**Table 1.** $^1H$- and $^{13}C$-NMR spectral of maslinic acid-2-O-β-D-glucose (dimethylsulfoxide-D6, 400 MHz).

| C | $^1H$ NMR (ppm) | $^{13}C$ NMR (ppm) |
|---|---|---|
| 1 | 1.93, 1H, dd, $J$ = 4.4, 13.0 Hz <br> 0.88, 1H, overlapped | 44.71 |
| 2 | 3.55, 1H, m | 79.44 |
| 3 | 2.91, 1H, d, $J$ = 9.2 Hz | 80.41 |
| 4 | - | 39.52 |
| 5 | 0.79, 1H, br d, $J$ = 11.3 Hz | 54.73 |
| 6 | 1.49, 1H, m <br> 1.33, 1H, m | 18.37 |
| 7 | 1.61, 1H, m <br> 1.43, 1H, m | 32.75 |
| 8 | - | 39.36 |
| 9 | 1.56, 1H, m | 47.50 |
| 10 | - | 38.11 |
| 11 | 1.85, 1H, m <br> 1.48, 1H, m | 23.12 |
| 12 | 5.17, 1H, br s | 121.67 |
| 13 | - | 144.58 |

**Table 1.** *Cont.*

| C | $^1$H NMR (ppm) | $^{13}$C NMR (ppm) |
|---|---|---|
| 14 | - | 41.85 |
| 15 | 1.68, 1H, m<br>0.98, 1H, m | 27.67 |
| 16 | 1.85, 2H, m | 23.56 |
| 17 | - | 45.93 |
| 18 | 2.76, 1H, $J$ = 3.6, 15.6 Hz | 41.32 |
| 19 | 1.60, 1H, m<br>1.05, 1H, m | 46.25 |
| 20 | - | 30.91 |
| 21 | 1.32, 1H, m<br>1.14, 1H, m | 33.89 |
| 22 | 1.43, 1H, m<br>1.25, 1H, m | 32.64 |
| 23 | 0.74, 3H, s | 17.75 |
| 24 | 0.96, 3H, s | 29.26 |
| 25 | 0.90, 3H, s | 16.62 |
| 26 | 0.72, 3H, s | 17.32 |
| 27 | 1.10, 3H, s | 26.15 |
| 28 | - | 179.29 |
| 29 | 0.88, 3H, s | 33.36 |
| 30 | 0.88, 3H, s | 23.87 |
| 1′ | 4.23, 1H, d, $J$ = 7.8 Hz | 102.82 |
| 2′ | 2.94, 1H, t, $J$ = 8.2 Hz | 73.80 |
| 3′ | 3.14, 1H, m | 77.00 |
| 4′ | 3.06, 1H, t, $J$ = 9.0 Hz | 70.41 |
| 5′ | 3.15, 1H, m | 77.18 |
| 6′ | 3.65, 1H, brd, $J$ = 13.6 Hz<br>3.43, 1H, dd, $J$ = 5.8, 13.6 Hz | 61.42 |

### 2.4. The Optimization of MA Glycosylation by UGTs

Specific enzymes can exert optimal catalytic activity at their optimum pH and temperature. Results have shown that the optimum pH range for YojK was 8.0−10.0, while the optimum pH value for YjiC and UGT109A3 was 8.0 for biotransformation of MA into MA-2-O-β-D-Glu (Figure 2A). Moreover, the results indicate that the optimum temperature for YojK and UGT109A3 glycosylation was 40 °C, and the optimum temperature for YjiC glycosylation was 45 °C (Figure 2B).

Since specific metal ions affect the catalytic activity of UGTs [12,16], the influence of different metal ions on MA biotransformation, which was catalyzed by YojK, YjiC, and UGT109A3, was explored. YojK showed efficient glycosylation when $Mg^{2+}$, $Na^+$, $K^+$, and $Li^+$ were present, while the activity of YojK was decreased in the presence of $Ca^{2+}$, $Mn^{2+}$, and $Co^{2+}$, and strongly inhibited in the presence of $Cu^{2+}$, $Fe^{2+}$, $Zn^{2+}$, and $Al^{3+}$. However, YjiC and UGT109A3 showed efficient MA glycosylation when $Mg^{2+}$, $Na^+$, $K^+$, $Li^+$, $Ca^{2+}$, and $Mn^{2+}$ were present. While in the presence of $Co^{2+}$, $Cu^{2+}$, $Fe^{2+}$, $Zn^{2+}$, and $Al^{3+}$, the catalytic activity of YjiC and UGT109A3 were decreased. $Mg^{2+}$ is the common metal ion for efficient glycosylation of YojK, YjiC, and UGT109A3 (Figure 2C). The above results indicate that $Mg^{2+}$ is likely to be a cofactor which enhanced the catalytic activity of YojK, YjiC, and

UGT109A3 by interacting with their active sites. Notably, the catalytic efficiency of YojK for MA glycosylation can reach 99.8% at pH 9.0/40 °C in the presence of $Mg^{2+}$, the catalytic efficiency of YjiC for MA glycosylation can reach 94.6% at pH 8.0/45 °C in the presence of $Mg^{2+}$, and the catalytic efficiency of UGT109A3 for MA glycosylation can reach 98.3% at pH 8.0/40 °C in the presence of $Mg^{2+}$ (Figure 2C).

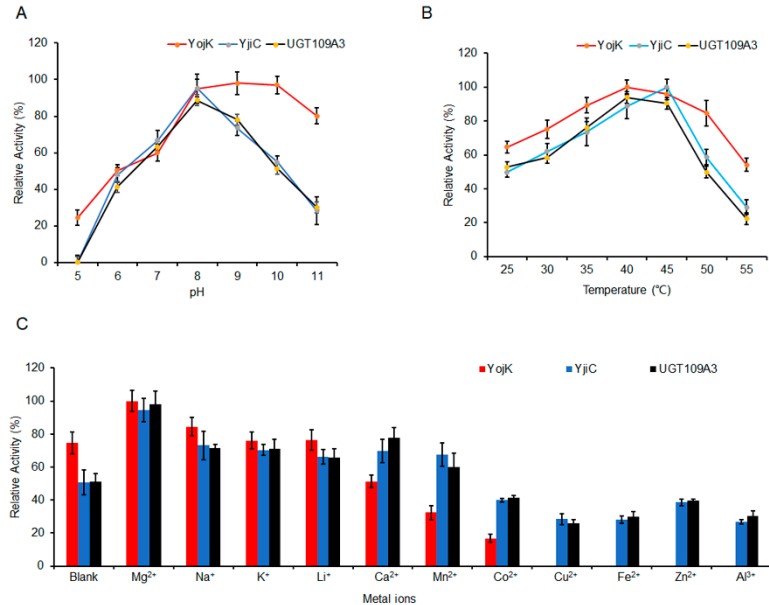

**Figure 2.** Effect of pH (**A**), temperature (**B**), and metal ions (**C**) on YojK, YjiC, and UGT109A3 activity for MA glycosylation.

The Michaelis constant (Km) value of YojK (0.23 mM) was lower than that of YjiC (0.48 mM) and UGT109A3 (0.52 mM), indicating that YojK has a higher affinity for MA (Table 2). In addition, the turnover value (Kcat) of YojK, YjiC, and UGT109A3 were 4.61 $s^{-1}$, 3.31 $s^{-1}$, and 3.78 $s^{-1}$, respectively (Table 2). This result indicated that YojK leads to the conversion of more MA to MA-2-O-β-D-Glu in one second. Furthermore, the result demonstrated that YojK is a more efficient enzyme to catalyze MA glycosylation thanf YjiC and UGT109A3.

**Table 2.** Kinetic parameters of UGTs (Mean ± SD, *n* = 3).

| Enzymes | Km (mM) | Kcat ($s^{-1}$) |
|---|---|---|
| YojK | 0.23 ± 0.02 | 4.61 ± 0.35 |
| YjiC | 0.48 ± 0.05 | 3.31 ± 0.12 |
| UGT109A3 | 0.52 ± 0.03 | 3.78 ± 0.13 |

*2.5. Exploring the Essential Residues of YojK, YjiC, and UGT109A3*

Of the three, the crystal structure of YjiC is the only one known to be resolved [17,18]. Molecular docking of YjiC with ligand MA and UDPG was performed by AutoDock Vina. Molecular docking results show that H16, D106, S277, Q280, S298, H293, E317, and Q318 of YjiC may be crucial amino acid residues in catalyzing MA glycosylation (Figure 3A). Notably, sequence alignment analysis reveals that these sites are considerably conserved among YojK, YjiC, and UGT109A3 (Figure 3B). Furthermore, the amino acids were replaced with alanine (A), as shown in Figure 3B. To assess the effects of the amino acid mutations on the function of YjiC, UGT109A3, and YojK, respectively, the biotransformation of MA into MA-2-O-β-D-Glu was measured at a fixed substrate concentration. As shown in Figure 3C–E, the MA conversion rate of YjiC-mut, YojK-mut, and UGT109A3-mut is less than 20%, compared to YjiC-wt, YojK-wt, and UGT109A3-wt, respectively. The results

showed that the mutation of the above amino acids resulted in a drastic decrease in the catalytic activity of YojK, YjiC, and UGT109A3. Previous literature revealed that H16 of YjiC was responsible for the deprotonation of the hydroxyl moiety on the sugar acceptor, and D106 could increase the proton-accepting capacity of H16 [17]. H16 and D106 of YjiC, H14 and D107 of YojK, H16 and D106 of UGT109A3 were possibly also involved in proton transfer of C2-hydroxyl on MA. The nucleotide base binding sites (S277-Q280), pyrophosphate binding loop sites (H293-S298), and D/E-Q motif (E317 and Q318) of YjiC were sugar donor binding sites, which played key roles in binding to UDPG [18]. Y288-Q291, H304-S309, D328, and Q329 of YojK, S277-Q280, H293-S298, E317, and Q318 of UGT109A3 may also key binding sites of UDPG. Moreover, these essential residues of YojK, YjiC, and UGT109A3 could be employed to predict the reactivity of other terpenoid substrates.

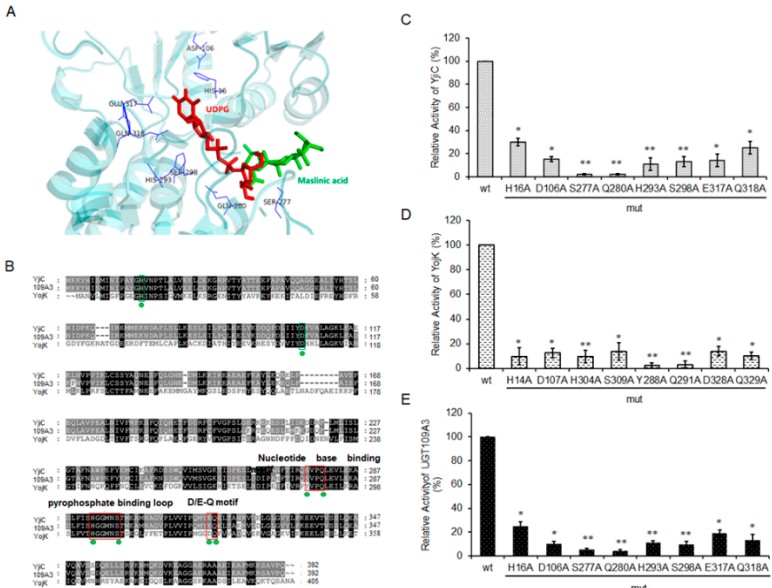

**Figure 3.** The key amino acids of YojK, YjiC, and UGT109A3. (**A**) Molecular docking of YjiC showing amino acid interactions with MA and UDPG. (**B**) Amino acids sequence alignment of YojK, YjiC, and UGT109A3. The green frame indicates the active sites His-Asp. The red frames indicate nucleotide base binding motif, pyrophosphate binding loop, and D/E-Q motif interacts with the UDP moiety. The green dots indicate the amino acids for site-directed mutagenesis. Effect of YjiC-mut (**C**), YojK-mut (**D**), or UGT109A3-mut (**E**) activity for MA glycosylation. * $p < 0.05$, ** $p < 0.01$ vs. wt.

## 3. Materials and Methods

### 3.1. Chemicals and Reagents

Chemicals/reagents (supplier, city, country): MA (Sigma-Aldrich, Shanghai, China); UDP-Glucose (UDPG) (Macklin Biochemical, Shanghai, China); antibodies (Abcam, Cambridge, MA, USA). Anti-6×His Tag (ab9108), Goat Anti-Rabbit IgG (ab6721) antibodies.

### 3.2. Plasmid Construction and Heterologous Protein Expression

The plasmid construction of pET-28a-YojK [12], pET-28a-YjiC [18], and pET-28a-UGT109A3 [13] were performed as described previously. The heterologous protein expressions of YojK, YjiC, and UGT109A3 were detected by SDS-PAGE and Western blot.

### 3.3. Protein Purification

The soluble proteins (YojK, YjiC, and UGT109A3) were purified as described previously [12].

### 3.4. Enzymatic Assays for MA Glycosylation In Vitro

Glycosylation of MA was accomplished in 1 mL reaction mixtures containing 10 mM PBS buffer (pH 7.4), 0.2 mM MA, 1.2 mM UDPG, and 0.25 μg purified YojK, YjiC, or

UGT109A3 enzyme. Reaction mixtures were incubated at 37 °C for 30 min and then analyzed by High Performance Liquid Chromatography (HPLC) for MA glycosylation.

### 3.5. HPLC Analysis

The Agilent 1260 instrument (Agilent Technologies Inc., Santa Clara, CA, USA) was used for MA glycosylation chromatographic analysis. The chromatographic separations conditions were UV210 nm, 40 °C, reverse phase C18 column (5 mL; 250 × 4.6 mm; Shimadzu, Kyoto, Japan), mobile phase (methanol: 0.6% acetic acid = 80:20 *v/v*), flow rate (1 mL/min).

### 3.6. LC-MS and NMR Identification of the MA-Glycosides

The preparative LC3000 HPLC system (Chuang Xin Tongheng, Beijing, China) was used to purify MA-glycosides. The structure of the compound was confirmed with a liquid chromatograph-mass spectrometer (LC-MS) (Agilent 1260–6120, Agilent Technologies Inc., Santa Clara, CA, USA) and Nuclear Magnetic Resonance (NMR) (BRUKER AVIII 400MHz, Bruker Corp., Karlsruhe, Germany) analysis.

### 3.7. Determination of Solubility

The solubility of MA and MA-2-O-β-D-Glu were determined experimentally as described previously [19].

### 3.8. The Optimization of Enzymatic Glycosylation Reaction

The optimum pH, temperature, metal ions, and kinetic parameters of the YojK, YjiC, and UGT109A3 for MA glycosylation were explored as described previously [12].

### 3.9. Molecular Docking

Molecular docking of MA, UDPG, and YjiC was done by AutoDock Vina, PyMOL v1.7.4.5, Swiss Model, and chem3D 16.0 [20].

### 3.10. Site-Directed Mutagenesis

Site-directed mutagenesis of pET-28a-YojK, pET-28a-YjiC, and pET-28a-UGT109A3 was performed using a Quick-Change Site-Directed Mutagenesis kit (Agilent Technologies, Santa Clara, CA, USA), using the primers listed in Table S1.

### 3.11. Statistical Analyses

SPSS was used for statistical analysis.

## 4. Conclusions

In this work, a new MA glycosylation product, MA-2-O-β-D-Glu, was synthesized by UGTs (YojK, YjiC, and UGT109A3) from *B. subtilis* for the first time. MA-2-O-β-D-Glu has better solubility than that of MA. In addition, the catalytic efficiency of YojK, YjiC, and UGT109A3 for MA glycosylation can reach nearly 100% under optimal reaction conditions. At the same time, due to their promiscuity, these enzymes can be used for biotransformation between other triterpenoids and their glycosylated derivatives in pharmaceutical and industrial production.

**Supplementary Materials:** The following supporting information can be downloaded at: https://www.mdpi.com/article/10.3390/catal12080884/s1, Table S1: The primers of site-directed mutagenesis.

**Author Contributions:** Conceptualization, F.H. and K.X.; methodology, H.C.; validation, Y.L.; formal analysis, J.C.; investigation, J.C. and Y.Z.; resources, Y.Z.; data curation, Y.S.; writing—original draft preparation, F.H.; writing—review and editing, K.X.; visualization, Y.Y.; funding acquisition, F.H., Y.L., K.X. and H.C. All authors have read and agreed to the published version of the manuscript.

**Funding:** This research was funded by S&T program of Hebei (H2021209007, 21374301D, B2022105015), Key Laboratory of Hebei Province (SZX2020043), Program for Innovation Research Team (in Science and Technology) of Tangshan (20130202D) and the S&T Program of Tangshan (21130201C).

**Institutional Review Board Statement:** Not applicable.

**Informed Consent Statement:** Not applicable.

**Data Availability Statement:** Not applicable.

**Conflicts of Interest:** The authors declare no conflict of interest.

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
