# Peer review of "Novel Biotransformation of Maslinic Acid to MA-2-O-β-D-Glucoside by UDP-Glycosyltransferases from Bacillus subtilis"

_catalysts, doi:10.3390/catal12080884_

Round 1

Reviewer 1 Report

The article is relevant to the scope of the journal. The authors investigate three glycosyltransferases involved in conjugating a UDP-glucose molecule to a known triterpenoid MA. The whole premise for the manuscript hinges on the prediction that in doing so, solubility of MA is increased -- which is then computationally predicted, but no experimental evidence provided. The authors do more experiments to provide optimized conditions for the enzymes, but fail to provide the key experiments needed to show increased solubility of the substrate itself, which should be a pretty straight forward experiment to do.

Some key observations:

1.      Plenty of grammatical errors are highlighted.

2.      Figure 1 should be a schematic and not a figure.

3.      Figures 1-3 can be condensed into a singular figure, since these are standard confirmatory experiments.

4.      Solubility prediction by modeling and prediction is not a largely accepted parameter. The authors should provide additional experimental evidence to support this claim.

Author Response

please see the attach file

Reviewer 2 Report

The article entitled “Novel Biotransformation of Maslinic acid to MA-2-O-beta-D-Glucoside by UDP-Glycosyltransferases from Bacillus Subtilis” by Hu et. al. shows the efficient transformation of a pentacyclic terpenoid (maslinic acid) to its glycosylated form by the action of three previously described recombinant glycosyltransferases. The experiments presented in the article are well performed and shows the novelty of expanding the substrate specificity of described recombinant glycosyltransferases.

Nevertheless, the article lacks certain experiments and discussion in order to get it published in Catalyst.

Major points:

In Figure 2, the HPLC traces clearly show that YojK enzyme produced the product more efficiently than the other two enzymes at short reaction times. A kinetic characterization of the three enzymes must be included. In different parts of the article, it is discussed that the transformation occurs at 100 % when optimal conditions are applied but this is not shown, the isolated yields/times for the preparative reactions should be included in the experimental part.

The introduction of Glucose is claimed to occur in C2 of the maslinic acid, it is need to be discussed which signals in the NMR experiments support this affirmation.

The solubility was predicted to increase in the glycosylated terpenoid, could it be determined experimentally and included in the paper?

The residues involved in the catalysis were determined by site directed mutagenesis and molecular docking, could be this information employ to predict the reactivity of other terpenoid substrates?

A careful revision of the English should be done because certain sentences are difficult to understand and more meaningful discussion of the novelty and findings should be included.

Author Response

  1. In Figure 2, the HPLC traces clearly show that YojK enzyme produced the product more efficiently than the other two enzymes at short reaction times. A kinetic characterization of the three enzymes must be included. In different parts of the article, it is discussed that the transformation occurs at 100 % when optimal conditions are applied but this is not shown, the isolated yields/times for the preparative reactions should be included in the experimental part.

Answer: We have added a kinetic characterization of the three enzymes and the isolated yields for the preparative reactions to the manuscript.

2.The introduction of Glucose is claimed to occur in C2 of the maslinic acid, it is need to be discussed which signals in the NMR experiments support this affirmation.

Answer: We have added to the manuscript.

3.The solubility was predicted to increase in the glycosylated terpenoid, could it be determined experimentally and included in the paper?

Answer: We feel very sorry, as testing the solubility of the glycosylated terpenoid requires a large amount of purified product, we cannot determine experimentally.

4.The residues involved in the catalysis were determined by site directed mutagenesis and molecular docking, could be this information employ to predict the reactivity of other terpenoid substrates?

Answer: This information could employ to predict the reactivity of other terpenoid substrates.

5.A careful revision of the English should be done because certain sentences are difficult to understand and more meaningful discussion of the novelty and findings should be included.

Answer: We have a careful revision of the English. And we have discussed more meaningful discussion of the novelty and findings.

Round 2

Reviewer 1 Report

The authors claim to have presented solubility data on Page 5 of manuscript, which I cannot find in the revised manuscript. They have some kinetic data presented for the enzymes, and claim they show solubility data both in the rebuttal letter and the conlcusion -- but I cannot find it in a figure or table. They also address the same concern by Reviewer 1 by stating that this expriment cannot be done, so I am quite confused by their approach.

Reviewer 2 Report

Accept the article in present form, 
